# Optimal Noise pursuit for Augmenting Text-to-Video Generation

## Abstract

Despite the remarkable progress in text-to-video generation, existing diffusion-based models often exhibit instability in terms of noise during inference. Specifically, when different noises are fed for the given text, these models produce videos that differ significantly in terms of both frame quality and temporal consistency. With this observation, we posit that there exists an optimal noise matched to each textual input; however, the widely adopted strategies of random noise sampling often fail to capture it. In this paper, we argue that the optimal noise can be approached through inverting the groundtruth video using the established noise-video mapping derived from the diffusion model. Nevertheless, the groundtruth video for the text prompt is not available during inference. To address this challenge, we propose to approximate the optimal noise via a search and inversion pipeline. Given a text prompt, we initially search for a video from a predefined candidate pool that closely relates to the text prompt. Subsequently, we invert the searched video into the noise space, which serves as an improved noise prompt for the textual input. In addition to addressing noise, we also observe that the text prompt with richer details often leads to higher-quality videos. Motivated by this, we further design a semantic-preserving rewriter to enrich the text prompt, where a reference-guided rewriting is devised for reasonable details compensation, and a denoising with a hybrid semantics strategy is proposed to preserve the semantic consistency. Extensive experiments on the WebVid-10M benchmark show that our proposed method can improve the text-to-video models with a clear margin, while introducing no optimization burden.

## 1 Introduction

Text-to-video generation has emerged as a valuable approach in automated video production, offering a human-friendly method with wide applications in diverse industries such as media, gaming, film, and television. Following the paradigm of conditional diffusion models (Ho et al., 2020; Song et al., 2020a;b; Dhariwal & Nichol, 2021), the text-to-video generation models typically sample a Gaussian noise in each inference, which is together with the text condition to synthesize the videos. However, our observations have indicated that different noises can yield significantly varied videos in terms of frame quality and temporal consistency. Upon closer examination, this observation highlights the existence of an optimal noise sample for a given text prompt, but the strategy of random sampling noise is hard to hit or get close to it every time. As a result, video quality varies, with higher quality achieved when the noise sample aligns closely to the optimal point, and poorer quality observed when it deviates further away.

Given this consideration, our objective in this paper is to approach the optimal noise for a given text prompt in order to consistently generate high-quality videos. Notably, the inversion and denoising procedures within the diffusion model establish a bidirectional mapping between the noise space and the video space. Based on the fact that the inversion of a sample can almost reconstruct itself through denoising (Mokady et al., 2023), we think that the inversion of the grundtruth video of the given text prompt should be close enough to the optimal noise. Nevertheless, the groundtruth video is not available during inference. To tackle this challenge, we propose to leverage the neighboring counterpart of the desired video to approximate the optimal noise.

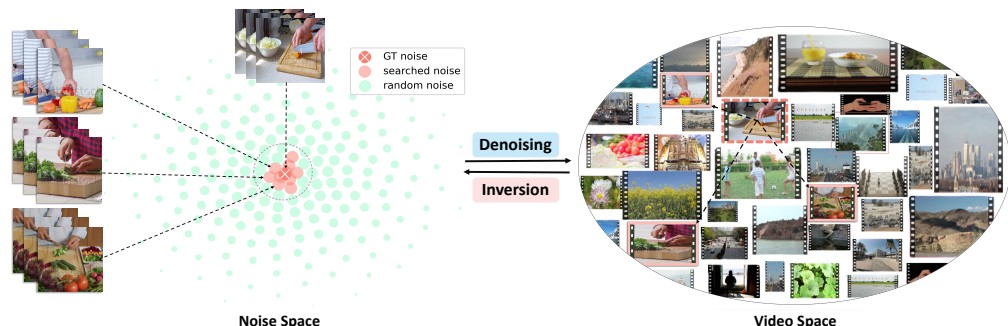

Figure 1: Illustration of our motivation. The trained denoising and inversion functions establish a bidirectional mapping between the video space and the noise space. Treating the inversion of the ground truth video ("GT noise") is the optimal noise, our objective is to approximate this optimal noise by inverting video neighbors. It is observed that similar videos converge to a confined region within the noise space, forming the theoretical basis for our optimal noise approximator.

To be specific, we first search a neighbor video for the text input, and then apply the inversion procedure on it to locate a point in the noise space as shown in Figure 1. Motivated by this concept, we initially assemble a pool of text-video pairs. For a given input text prompt, we select the video from the pair whose text is similar with the text prompt as the neighbor video. Subsequently, this chosen video is inverted into the noise space as an approximation for the optimal noise. Finally, the approximated noise is input to the forward procedure in the diffusion model, resulting in the synthesis of higher-quality videos. However, using the fixed noise for every text would lead to poor diversity, to remedy this issue, we further augment the noise initialization by introducing the random noise via Gaussian Mixture to maintain the diversity.

In addition to noise inputs, text prompts also play a pivotal role in influencing the quality of generated videos. Detailed descriptions tend to produce superior results, as demonstrated in (Fan et al., 2023; Witteveen & Andrews, 2022; Pavlichenko & Ustalov, 2023; Hao et al., 2022). To harness this characteristic, a straightforward approach is to enhance the text descriptions using large-scale language models (LLMs) such as ChatGPT (OpenAI, 2023) and Llama2 (Touvron et al., 2023). However, this naïve strategy presents two challenges: **(a)** The potential information gain through simple rewriting is often limited, resulting in suboptimal improvements to the text prompts. **(b)** Without appropriate guidance, the introduction of unexpected content may lead to the generation of videos that deviate from the user's original intention.

For the first issue, we propose a Reference-Guided Rewriting (RGR) mechanism, RGR searches several texts as the reference for the rewriting, serving as the information pool to aid the LLMs to compensate reasonable details for input text. For the second challenge, we design a new inference strategy, named Denoising with Hybrid-Semantics (DHS), the rewritten text acts as the contextual condition in the early diffusion steps for quality enhancement, while the original text is introduced in later steps of the denoising process to align the semantics with the original text. By strategically including the original text, DHS ensures that the final video closely adheres to the original text, maintaining the intended narrative consistency.

Through advancements in enhancing the two inputs of text-to-video models, we have successfully developed an optimization-free and (diffusion) model-agnostic method. In summary, we make the following contributions in this paper: **1) An optimal noise approximator** that offers the noise initialization close to the optimal noise for each text prompt. **2) A semantic-preserving rewriter** characterised by the RGR and DHS to provide detail-rich text prompts while maintaining the semantics of the final video to keep consistent with the original text.

## 2 RELATED WORK

Text-to-video generation within open domains, using diffusion models, is an active and compelling research area. The prevailing approaches (Ho et al., 2022b; Singer et al., 2022; Ho et al., 2022a; Zhou et al., 2022; Blattmann et al., 2023; Wu et al., 2022; Ge et al., 2023; Luo et al., 2023) typically

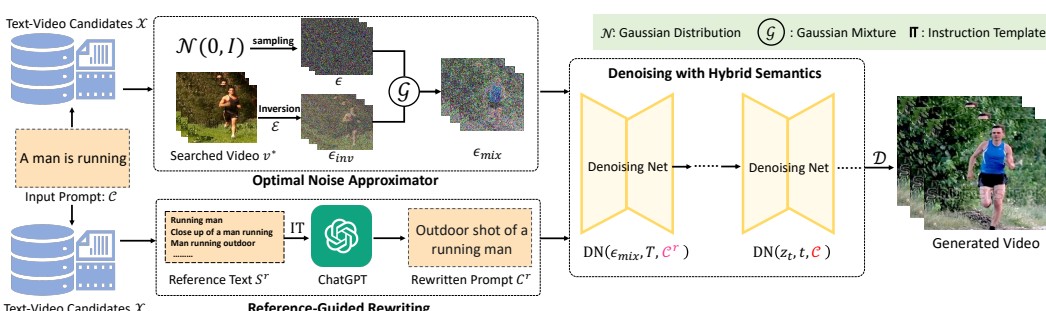

Figure 2: Illustration of our method. Given a trained text-to-video model, we enhance it by improving the two types of inputs: the noise and the text prompt. The optimal noise approximator targets to approach the optimal noise for the text prompt, while the semantic-preserving rewriter, formed by the reference-guided rewriting and the denoising with hybrid semantics, improves the text prompt by providing more details.

extend text-to-image models to facilitate video generation. This strategy leverages the knowledge from text-to-image models. Broadly, existing methods enhance text-to-video generation through two primary avenues: architecture-oriented optimization (Ho et al., 2022b; Singer et al., 2022; Ho et al., 2022a) and noise-oriented optimization (Luo et al., 2023; Ge et al., 2023). Architecture-oriented optimization typically designs modules for temporal relation enhancement. For example, VideoFactory (Wang et al., 2023a) proposes a novel spatiotemporal cross-attention to reinforce the interaction between spatial and temporal features. To improve the temporal consistency of the generated video, Tune-a-video (Wu et al., 2022) extends the spatial self-attention to the spatiotemporal attention by deriving key and value features from the first frame and the former frame. While noise-oriented optimization seeks to provide precise noise initialization to maintain video fluency more effectively. VideoFusion (Luo et al., 2023) decomposes the independent noise of individual images into a base noise and a residual noise, where all frames share the same base noise. For a similar motivation, Preserve Your Own Correlation (Ge et al., 2023) design two diffusion noise priors, where all video frames share a base noise like VideoFusion in the mixed noise model and the progressive noise model constructs the current frame noise based on the former frame noise in a progressive way. Instead of designing the noise empirically, this work targets to directly approximate the optimal noise for the text prompt. What's more, we also enrich the text prompts via the LLMs. Notably, the proposed modules are optimization-free and model-agnostic, which are also important features that distinguish our method from existing models.

## 3 METHODOLOGY

Figure2 depicts our framework, both ONA and SPR are seamlessly integrated without training. Given a trained text-to-video model and the text prompt, ONA finds a neighboring video and inverts it to noise space, combined with random noise for the noise input. SPR locates textual references for rewriting and performs denoising with hybrid semantics to preserve the original semantics.

### 3.1 DDIMs REVISIT

Before diving into the details of our method, we first revisit two crucial modules in Denoising Diffusion Implicit Models (DDIMs) (Song et al., 2020a), *i.e.,* the Denoising and the Inversion procedures, to ease the understanding of our method.

**Denoising** in diffusion model builds a mapping from normal noise to the samples (image, video *etc*). Compared with Denoising Diffusion Probabilistic Models (DDPMs) (Ho et al., 2020), DDIMs (Song et al., 2020a) are more efficient, allowing for the use of a smaller number of steps to synthesize a sample $z_0$ from a Gaussian noise $z_T$ using follow the iterative procedure:

$$z_{t-1} = \sqrt{\alpha_{t-1}} \times (\frac{z_t}{\sqrt{\alpha_t}} + (\sqrt{\frac{1-\alpha_{t-1}}{\alpha_{t-1}}} - \sqrt{\frac{1-\alpha_t}{\alpha_t}})\epsilon_\theta(z_t, t, c)) \tag{1}$$

where $\alpha_t = \prod_{t=1}^{T}(1 - \beta_t)$, $\beta_t$ is a predefined hyperparameter in DDPMs, $c$ is the condition to control the generation, $\epsilon_\theta$ is a noise prediction network. We define function $\mathrm{DN}(z_t, t, c)$ as the right term of the above equation to simplify the subsequent description: $z_{t-1} = \mathrm{DN}(z_t, t, c)$.

**Inversion** defines a projection from data space to noise space, which is a symmetrical mapping of denoising. Given a sufficiently large $T$, Eq.1 approaches an ordinary differential equation (ODE). With the assumption that the ODE process can be reversed in the limit of small steps (Mokady et al., 2023), we can acquire a noise from a data $z_0$:

$$z_{t+1} = \sqrt{\alpha_{t+1}} \times \left( \frac{z_t}{\sqrt{\alpha_t}} + \left( \sqrt{\frac{1 - \alpha_{t+1}}{\alpha_{t+1}}} - \sqrt{\frac{1 - \alpha_t}{\alpha_t}} \right) \epsilon_\theta(z_t, t, c) \right) \tag{2}$$

Analogously, we define the right term of the above equation as $\mathrm{INV}(z_t, t, c)$ to convince the subsequent elaboration: $z_{t+1} = \mathrm{INV}(z_t, t, c)$.

## 3.2 Optimal Noise Approximator

ONA (Optimal Noise Approximator) first searches a video as the inversion source and then applies the DDIM inversion to approach the optimal noise.

**Video Retrieval** seeks to find a video $v*$ that aligns with the text prompt in semantics. We first prepare a pool of $N$ text-video pairs $\mathcal{X} = \{S_i, V_i\}_{i=1}^{N}$, where $S$ and $V$ represent text and video, respectively. Given a text prompt $\mathcal{C}$, we initially estimate the similarity between the text prompt and the text within $\mathcal{X}$ and then select the video whose text counterpart shares the highest similarity with the prompt as the inversion source:

$$v* = \{V_i | \arg\max_i \{\mathrm{sim}(E_t(\mathcal{C}), E_t(S_i)) | \{S_i, V_i\} \in \mathcal{X}\}\} \tag{3}$$

where $\mathrm{sim}(\cdot, \cdot)$ is the cosine similarity, and $E_t$ can be a off-the-shelf semantic language model like (Reimers & Gurevych, 2019; Devlin et al., 2019), to extract the text feature. The reasons we anchor on the text-similarity for the video selection are three-fold: first, text similarity is a more reliable clue than the text-video relevance; second, the computation efficiency for text feature extraction is superior; third, due to the available text-video dataset like WebVid-10M (Bain et al., 2021), InvernVid (Wang et al., 2023b), collecting the text-video pairs is not effort-intensive as well.

**Guided Noise Inversion** module takes the searched $v*$ as the inversion source to approximate the optimal noise. Formally, we follow the efficient latent diffusion architecture (Rombach et al., 2021), where the raw video $v*$ is first encoded via the encoder of VQ-VAE (Razavi et al., 2019) and then the inversion is recurrently acted on the latent feature with $T$ steps: $\epsilon_{\mathrm{inv}} = \mathrm{INV}(\mathcal{E}(v*), t, \varnothing)|_{t=0}^{T-1}$, where $\epsilon_{\mathrm{inv}}$ is the inverted noise, $\varnothing$ represents the empty text and $\mathcal{E}$ refers to the VQ-VAE encoder.

As shown in Figure 1, this simple strategy can help locate a point close to the optimal noise. On the other hand, the prompts that share the inversion source video will use the same noise; especially, multiple inferences of the same prompt will yield the same result, severely sacrificing the diversity. In response to this shortcoming, we introduce the Gaussian noise and integrate it with the inverted noise $\epsilon_{\mathrm{inv}}$ via Gaussian mixture to maintain the randomness:

$$\epsilon_{\mathrm{mix}}(v*, \eta) = \frac{1}{\sqrt{1 + \eta^2}} \cdot \epsilon + \frac{\eta}{\sqrt{1 + \eta^2}} \cdot \epsilon_{\mathrm{inv}} \tag{4}$$

where $\epsilon \sim \mathcal{N}(0, \mathbf{I})$ is the Gaussian noise, $\eta$ is the hyperparameter to balance the noises.

**Video Synthesize with Improved Noise.** Given the improved noise $\epsilon_{\mathrm{mix}}$ and text prompt $\mathcal{C}$, the latent code $z_0$ of the video is calculated by the following formula:

$$z_{t-1} = \begin{cases} \mathrm{DN}(\epsilon_{\mathrm{mix}}(v*, \eta), t, \mathcal{C}) & \text{if } t = T \\ \mathrm{DN}(z_t, t, \mathcal{C}) & \text{if } 0 < t \leq T - 1, \end{cases} \tag{5}$$

where $T$ is the total DDIM sampling steps.

## 3.3 Semantic-Preserving Rewriter

Semantic-Preserving Rewriter (SPR) targets to improve another crucial input for text-to-video generation, *i.e.,* the text prompt, which is motivated by the observation that more detailed descriptions

yield improved results (Fan et al., 2023; Witteveen & Andrews, 2022; Pavlichenko & Ustalov, 2023; Hao et al., 2022). In particular, SPR comprises Reference-Guided Rewriting (RGR) and Denoising with Hybrid Semantics (DHS).

**Reference-Guided Rewriting** aims to provide descriptions to guide the rewriting, allowing the LLMs to "imagine" reasonable textual details. In particular, we first use the Sentence-BERT model (Reimers & Gurevych, 2019) to encode the text prompt. Consider a text prompt $\mathcal{C}$, its top-$k$ references are selected using the cosine similarity:

$$\{S_i^r\}_1^k = \arg \text{top} \, k\{\text{sim}(E_t(\mathcal{C}), E_t(S_i))|S_i \in \mathcal{X}\}, \tag{6}$$

where $\text{sim}(\cdot, \cdot)$ is the cosine similarity, and $E_t$ is the text encoder of the Sentence-BERT model.

Subsequently, we integrate the references $\{S_i^r\}_1^k$ and the text prompt $\mathcal{C}$ into a designed instruction template $\text{IT}$, which is fed into a LLM to rewrite the prompt:

$$\mathcal{C}^r = \text{LLM}(\text{IT}\{\{S_i^r\}_1^k, \mathcal{C}\}), \tag{7}$$

**Denoising with Hybrid Semantics.** Reference sentences provide valuable guidance for the reasonable details compensation, however, we found this strategy cannot perfectly maintain the original semantics of the text prompt due to the excellent association ability of LLMs. To remedy this issue, we propose to introduce the original text prompt into the denoising process. Specifically, we apply the rewritten sentence as the condition in the early stage to boost the content quality, while the original text prompt is employed in the latter denoising steps to pull the semantics close to the original prompt. As a result, we evolve the video synthesis of Eq. 5 as follows:

$$z_{i-1} = \begin{cases} \text{DN}(\epsilon_{\text{mix}}(v*, \eta), t, \mathcal{C}^r), & \text{if } t = T \\ \text{DN}(z_t, t, \mathcal{C}^r) & \text{if } T - m < t \leq T - 1 \\ \text{DN}(z_t, t, \mathcal{C}) & \text{if } 0 < t \leq T - m, \end{cases} \tag{8}$$

where $m = \lfloor T \times \gamma \rfloor, \gamma \in (0, 1)$. $m$ indicates how many steps of rewritten text is performed, $\lfloor \cdot \rfloor$ is the floor operation. Following the latent diffusion model (Rombach et al., 2021), we finally synthesize the video by feeding the final latent feature $z_0$ into the VQ-VAE decoder: $\hat{v} = \mathcal{D}(z_0)$.

## 4 EXPERIMENT

### 4.1 STANDARD EVALUATION SETUP.

During our practice, we found that the evaluation settings in existing text-to-video works are either varied or ambiguous, posing the risk of unfair comparison. In light of this issue, we present our detailed evaluation configuration and hope to standardize the future evaluation of text-to-video generation models.

**Datasets & Metrics.** We follow the previous works (Ho et al., 2022b; Singer et al., 2022; Zhou et al., 2022; Blattmann et al., 2023; Ge et al., 2023; Wang et al., 2023a) and adopt the widely-used **MSR-VTT** (Xu et al., 2016) and **UCF101** (Soomro et al., 2012) datasets for performance evaluation.

*MSR-VTT* provides 2,990 video clips for testing, each accompanied by around 20 captions. To evaluate the performance, we randomly select one caption per video to create a set of 2,990 text-video pairs for performance evaluation. Evaluation is conducted using FID (Heusel et al., 2017)[1] and CLIP-FID (Kynkäänniemi et al., 2022)[2] metrics to assess video quality, along with CLIP-SIM (Wu et al., 2021)[3] metric to measure semantic consistency between videos and text prompts.

*UCF101* contains 13,320 video clips representing 101 human action categories. We evaluate performance using 3,783 test videos. As there are no captions, we use the text prompts from PYoCo (Ge et al., 2023) for video generation. To ensure fair comparisons, we employ IS (Salimans et al., 2016) and FVD (Unterthiner et al., 2018; Yan et al., 2021) metrics on UCF101. Video IS (Singer et al., 2022; Hong et al., 2022) uses C3D(Tran et al., 2015) pretrained on UCF101 as the feature

---

[1]FID: https://github.com/GaParmar/clean-fid

[2]CLIP-FID: https://github.com/GaParmar/clean-fid

[3]CLIPSIM: https://github.com/openai/CLIP

extractor[4], while FVD utilizes I3D (Vadis et al.) pretrained on Kinetics-400 (Kay et al., 2017) for video feature encoding[5].

**Resolution.** Following previous works (Singer et al., 2022; Zhou et al., 2022; Ge et al., 2023; He et al., 2022), we generate videos of size $16 \times 256 \times 256$ for performance evaluation. This choice is motivated by two main factors: First, the training dataset WebVid-10M (Bain et al., 2021) for most text-to-video models typically consists of videos with a resolution of approximately 360P; Second, both the MSR-VTT and UCF101 datasets consist of videos with a resolution of around 240P. Consequently, $256 \times 256$ ensures a consistent and comparable evaluation setting, allowing for a reliable comparison.

**Sampling Strategy.** For FID and CLIP-FID, we sample 14950 images from 2990 real MSR-VTT videos and generated videos respectively. Concretely, we sample 5 images for each video in both the real and generated videos. For the real video, sample an image every 12 frames. For the generated video, sample an image every 4 frames. Regarding CLIP-SMI, we calculate the clip similarity between each image of the generated video and the corresponding prompt and then average it. All 3783 UCF101 test videos are used to calculate FVD and IS. For the real video, we sample an image every 5 frames, for a total of 16 frames, to calculate FVD.

## 4.2 IMPLEMENTATION DETAILS

We sample 100k image-text pairs from WebVid-10M to form our pre-defined candidate pool, *i.e.,* $N = 100k$. In Guided Noise Inversion, we search for the most similar video from the pool for each text prompt according to Eq. 3. Sentence-BERT is taken to extract text features[6]. In Semantic-Preserving Rewriter, we pick the top-5 ($k = 5$) reference sentences for the text prompt and adopt chatGPT (OpenAI, 2023) to perform rewriting, which imitates the adjectives, adverbs, or sentence patterns of these 5 sentences. The instruction template IT is set as *"Let me give you 5 examples:* [Ref1],[Ref2],[Ref3],[Ref4],[Ref5], *rewrite the sentence* [Input Text Prompt] *without changing the meaning of the original sentence to a maximum of 20 words, imitating/combining the adjectives, adverbs or sentence patterns from the 5 examples above"*.

The proposed modules can benefit many trained text-to-video models. To support this claim, we equip the proposed modules in the ModelScope (Luo et al., 2023)[7] and our trained SCVideo to study the performance gain. ModelScope is one of the stat-of-the-art diffusion-based models for text-to-video generation, we take it as a solid baseline to study the effectiveness of our modules. SCVideo is our designed model extended from text-to-image models by equipping temporal modules, including temporal convolution, Sparse Causal (SC) Attention, and temporal Attention. The architecture and training details of SCVideo, which are not the primary focus of this paper, will be provided in the Appendix. The hyperparameter of our method for our SCVideo on MSR-VTT and UCF101 is $\eta = 0.5$, $\gamma = 0.1$ and $\eta = 0.5$, $\gamma = 1.0$, respectively. In contrast, we choose $\eta = 0.2$, $\gamma = 0.04$ and $\eta = 0.2$, $\gamma = 1.0$ on MSR-VTT and UCF101 for ModelScope.

## 4.3 QUANTITATIVE RESULTS.

Table 1 reports the main results on two benchmarks, we can observe that our ONA and SPR can bring consistent performance improvement of two models on both datasets. Particularly, ONA and SPR can improve the FID of ModelScope from 45.378 to 43.092 and 43.585, respectively. Equipping both modules to make performance a step further, achieving a 42.755 FID. Furthermore, our modules also maintain a good semantic consistency, 0.296 *vs* 0.3. On UCF101 dataset, ONA shows great effectiveness, boosts the FVD of ModelScope from 774.14 to 607.11, and IS is also clearly improved. The contributions on our SCVideo are more remarkable, we can harvest an improvement of 5.429 FID on MSR-VTT and 209.31 of FVD on UCF101. These results well validate the effectiveness of our ONA and SPR. Figure 3 shows two groups of qualitative results, from which we can intuitively observe the effectiveness of our proposed modules.

---

[4]IS: https://github.com/pfnet-research/tgan2

[5]FVD: https://github.com/wilson1yan/VideoGPT

[6]Sentence-BERT: https://huggingface.co/sentence-transformers/paraphrase-multilingual-MiniLM-L12-v2

[7]ModelScope: https://huggingface.co/damo-vilab/text-to-video-ms-1.7b

Table 1: **Quantitative results** on MSR-VTT and UFC101 datasets, our ONA and SPR improve the performance with a clear margin, where ModelScope + ONA means ModelScope with ONA equipped, and **Bold** highlights the best performance

| | MSR-VTT | | | UCF101 | |
|---|---|---|---|---|---|
| | FID (inception v3)↓ | CLIP-FID ↓ | CLIPSIM ↑ | FVD ↓ | IS ↑ |
| ModelScope | 45.378 | 13.677 | 0.296 | 774.14 | 32.337 |
| ModelScope + ONA | 43.092 | **13.572** | 0.299 | 607.11 | **38.992** |
| ModelScope + SPR | 43.585 | **13.572** | 0.297 | 684.9 | 33.136 |
| ModelScope + ONA + SPR | **42.755** | 13.674 | **0.300** | **566.68** | 38.190 |
| SCVideo | 48.331 | 14.802 | 0.283 | 750.74 | 23.224 |
| SCVideo + ONA | 43.585 | 14.287 | **0.289** | 558.51 | 31.538 |
| SCVideo + SPR | 43.760 | 14.291 | 0.284 | 712.88 | 25.648 |
| SCVideo + ONA + SPR | **42.902** | **14.155** | 0.288 | **541.43** | **33.245** |

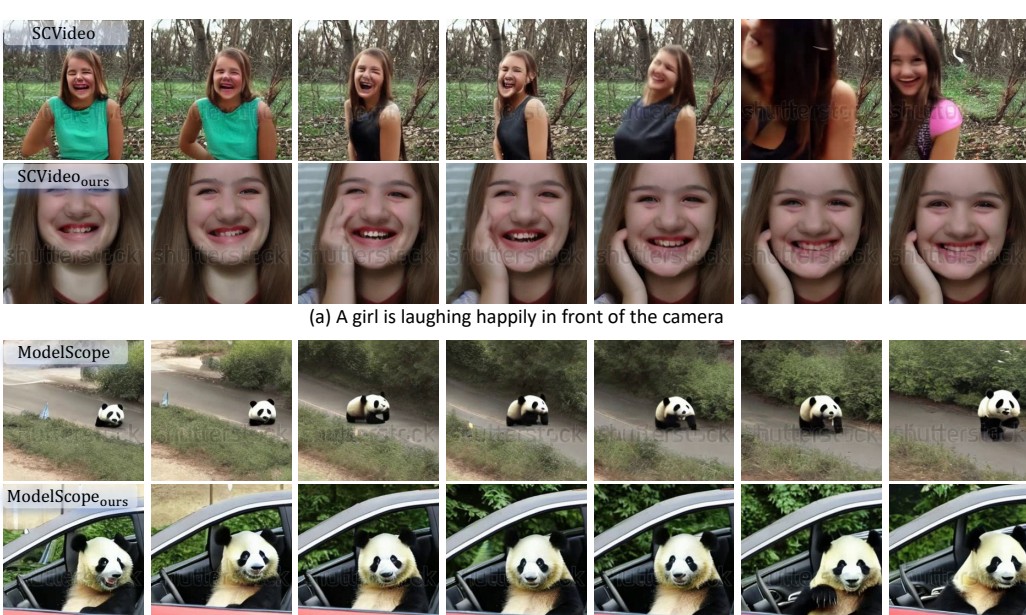

(a) A girl is laughing happily in front of the camera

(b) A panda is driving a car

Figure 3: **Qualitative results.** ModelScope$_{ours}$ means ModelScope with our ONA and SPR equipped, subfigure (a) and (b) show the results with SCVideo and ModelScope as backbones. Each group shares the same random noise for a fair comparison.

## 4.4 ABLATIONS

**Size of Candidate Pool.** To investigate the effect of candidate pool size, we randomly sample 10K, 100K, 1M, and 10M samples from the Webvid dataset to study the performance trend. Note that we report the final performance of applying both ONA and SPR (same below, unless otherwise stated). From Table 2a, we can observe that the larger the retrieval pool typically results in better performance. For example, enlarging the pool size from 10K to 10M can promote the FID on MSR-VTT from 43.236 to 42.072. Notably, although our default set is $N =$100K, reducing the scale to 10K does not severely hinder the performance. We take 100K to pursue a better trade-off between the scale of the candidate pool and the performance.

**Proportion of Guided Noise in ONA.** Hyperparameter $\eta$ in Eq. 4 determines the final composition of the noise, this part discusses the performance impact of how to hybridze the two types of noises by varying $\eta$. The results are reported in Figure 4, We can observe a clear tendency from the figure, that using the pure random noise ($\eta = 0$) or guided noise ($\eta = \infty$) can not yield satisfactory performance. Instead, an appropriate fusion of the two noise types is a more effective manner, we can harvest the best performance when $\eta = 0.5$.

Table 2: **Ablation experiments**. SCVideo is taken as the baseline to study the key hyperparameters in our ONA and SPR.

<table>
<tr><td colspan="6">(a) Size of candidate pool.</td><td colspan="6">(b) Proportion of Rewritten Text in DHS.</td></tr>
</table>

| | **MSR-VTT** | | | **UCF101** | | | | **MSR-VTT** | | | **UCF101** | |
|---|---|---|---|---|---|---|---|---|---|---|---|---|
| | FID↓ | CLIP-FID↓ | CLIPSIM↑ | FVD↓ | IS↑ | | | FID↓ | CLIP-FID↓ | CLIPSIM↑ | FVD↓ | IS↑ |
| $N=$ 10K | 43.236 | 14.352 | 0.288 | 591.79 | 31.735 | | $\gamma=0$ | 48.331 | 14.802 | 0.283 | 750.74 | 23.224 |
| $N=$ 100K | 43.585 | 14.287 | **0.289** | **558.51** | 31.538 | | $\gamma=0.1$ | 43.760 | **14.291** | **0.284** | 769.35 | 25.006 |
| $N=$ 1M | 42.587 | 14.334 | 0.288 | 620.29 | 33.047 | | $\gamma=0.2$ | 43.170 | 14.384 | 0.283 | 731.48 | 25.508 |
| $N=$ 10M | **42.072** | **14.285** | 0.289 | 571.27 | **33.232** | | $\gamma=0.5$ | **42.992** | 14.755 | 0.278 | 729.23 | 25.453 |
| | | | | | | | $\gamma=1.0$ | 43.531 | 15.484 | 0.275 | **712.88** | **25.648** |

<table>
<tr><td colspan="6">(c) Number of reference text in SPR.</td><td colspan="6">(d) Performance of different rewriting engines.</td></tr>
</table>

| | **MSR-VTT** | | | **UCF101** | | | | **MSR-VTT** | | | **UCF101** | |
|---|---|---|---|---|---|---|---|---|---|---|---|---|
| | FID↓ | CLIP-FID↓ | CLIPSIM↑ | FVD↓ | IS↑ | | | FID↓ | CLIP-FID↓ | CLIPSIM↑ | FVD↓ | IS↑ |
| K=0 | 45.435 | 15.504 | **0.278** | 808.16 | 23.849 | | Llama2-7B | 43.811 | 14.395 | 0.287 | 551.87 | 31.305 |
| K=1 | 45.410 | 15.386 | 0.273 | 745.07 | 24.604 | | ChatGPT | **42.901** | **14.155** | **0.288** | **541.43** | **33.245** |
| K=2 | 44.458 | 15.385 | 0.275 | 824.99 | 25.361 | | | | | | | |
| K=5 | 43.531 | 15.484 | 0.275 | **712.88** | 25.648 | | | | | | | |
| K=10 | **43.125** | **15.257** | 0.277 | 755.55 | **26.598** | | | | | | | |

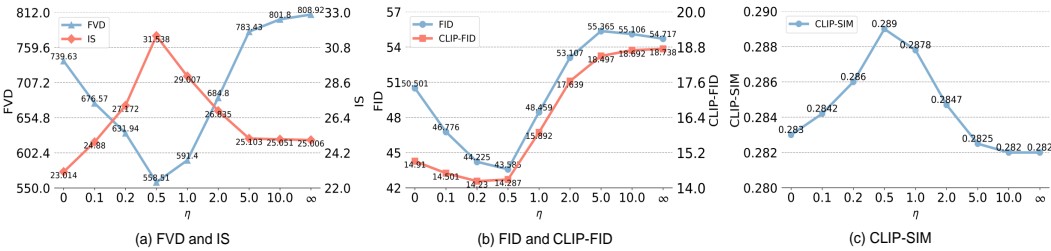

(a) FVD and IS  (b) FID and CLIP-FID  (c) CLIP-SIM

Figure 4: **Ablation study** on the hyperparameter $\eta$ in ONA, the case of $\eta = 0.5$ achieves the best performance on average (SCVideo serves as the baseline).

**Number of Dominated Steps of Rewritten Text in DHS.** The hyperparameter $\gamma$ controls how many steps of the rewritten text are performed in DHS. We vary $\gamma$ in this part and study its effect on performance, and do not equip the ONA to observe a clearer trend. As shown in Table 2b, the rewritten sentence ($\gamma = 1$) can boost the quality compared to the raw prompt. Specifically, compared with the baseline($\gamma = 0$ without rewritten text), applying the rewritten sentence reduces FID from 48.331 to 43.531 and promotes the IS score from 23.224 to 25.648. However, the CLIP-SIM drops from 0.283 to 0.275, indicating the semantic consistency is deteriorated. Our DHS strategy can remedy the issue, for example, introducing the original text in the last 90% denoising steps ($\gamma = 0.1$) can boost the CLIP-SIM from 0.275 to 0.284.

**Number of Reference Text for Rewriting.** By default, we employ 5 reference sentences for the text rewriting, this subsection examines the impact of varying the number of references for text rewriting. Table 2c presents the performance results across different reference numbers, highlighting the positive impact of incorporating references (we focus solely on the rewriting strategy, excluding the ONA and DHS techniques). For instance, the FID score is significantly improved from 45.435 without references to 43.531 with five references. Despite consistent improvements in other metrics, CLIP-SIM shows a decline due to the introduction of new content or objects. However, this issue can be effectively mitigated by our DHS mechanism, as evidenced in Table 1 and Table 2b.

**Effect of Large Language Models (LLMs).** ChatGPT is taken as the rewriting engine due to its excellent performance, we also present a discussion regarding the LLMs to further validate our methods. Table 2d compares the results of using ChatGPT (GPT-3.5-Turbo) and Llama2-7B in SPR. The findings indicate that ChatGPT consistently outperforms Llama2-7B across five key metrics, exhibiting a substantial advantage. Moreover, chatGPT also shows a higher level of intelligence in our practice, as it does not need regularization to filter out irrelevant content.

**Can ONA and SPR Benefit Image Generation?** To answer this question, we augment the Stable Diffusion (SD) 1.5 (Rombach et al., 2021) and Stable Diffusion-XL (SD-XL) (Podell et al., 2023) with our ONA and SPR and evaluate their performance on MS-COCO test set (Lin et al., 2014). We employ the test set of Flickr30k (Young et al., 2014) as the candidate pool instead of the MS-

Table 3: **Quantitative results** for text-to-image generation, the results are evaluated on MS-COCO dataset

| Backbones | **Stable Diffusion 1.5** | | | | **Stable Diffusion-XL** | | | |
|---|---|---|---|---|---|---|---|---|
| Metrics | SD 1.5 | + ONA | + SPR | + ONA & SPR | SD-XL | + ONA | + SPR | + ONA & SPR |
| FID (inception v3)↓ | 24.824 | 24.641 | 23.861 | **23.623** | 25.608 | 24.129 | 25.307 | **24.011** |
| CLIP-FID ↓ | 13.545 | 13.352 | 13.212 | **13.194** | 14.940 | 12.885 | 14.717 | **12.866** |
| CLIPSIM ↑ | 0.324 | **0.325** | 0.320 | 0.320 | 0.336 | **0.337** | 0.330 | 0.331 |

COCO training set to verify the generalization of our methods, and $\eta$ and $\gamma$ are set as 0.05 and 0.4, the other hyperparameters remain the same as video experiments. Table 3 compares the results, we have the following observations: (a) ONA yields a more substantial performance improvement for SD-XL compared to SD 1.5. This discrepancy arises from the fact that SD-XL samples noise from a larger space (128×128) in comparison to SD 1.5 (64×64), thereby rendering the task of achieving or approximating optimal noise more challenging for SD-XL. Consequently, ONA can leverage its strengths more effectively in the case of SD-XL. Conversely, the noise space of SD 1.5 is relatively constrained, and the extensive training with a voluminous dataset has effectively aligned the noise and image spaces. Consequently, the efficacy of ONA is less pronounced in this context. (b) SPR exhibits superior performance on SD 1.5 in contrast to SD-XL. This phenomenon can be attributed to SD-XL training a more potent text encoder, which is adept at capturing finer textual details.

## 4.5 LIMITATIONS

Our method has two-fold blemishes: (a) Lower inference efficiency. To approach the optimal noise and get a detail-richer text prompt, we introduce more processing steps compared to the vanilla diffusion models, as a result, the inference time is inevitably increased. Table 4 gives the detailed comparison, where the time is tested on NVIDIA A100 GPU with our SCVideo and averaged on 3000 samples. We can find the time for retrieval is negligible, and the feature extraction costs over 80% of addi-

Table 4: **Inference Time Ablation**.

| Modules | | **time (ms)** |
|---|---|---|
| ONA&SPR | Feature Extraction | 31,109 |
| ONA | Video Retrieval | **0.746** |
| | Inversion | **5,532** |
| SPR | Text Retrieval | **4.271** |
| | Rewriting | **1,531** |
| SCVideo | Denoising | **8,481** |

tional time. In practice, we can extract feature offline, as a result, the extra time is around 7 seconds. (b) Additional storage. Our method also requires more space to store the candidate pool. Original videos are center-cropped and resized to 256x256 for direct use. Since there is no need to store all frames, we sample an image every 8 frames and save the sampled video in MP4 format. Thus, the 100K video-text pairs occupy 45G disk space. However, as shown in Table 2a, 10K text-video pairs can also achieve satisfactory performance. To remedy these weaknesses, we will try to parameterize the noise approximator and text rewriter to evolve the proposed ONA and SPR in the future, such that the framework can get rid of the limitation of the search pool during inference.

## 5 CONCLUSION

This study presents an optimization-free, model-agnostic approach for enhancing text-to-video generation. Our method focuses on improving two crucial inputs: the noise and the text prompt. To approximate the optimal noise for a given text prompt, we propose an optimal noise approximator. This module involves a two-stage process, starting with the search for a video neighbor closely related to the text prompt. Subsequently, we perform DDIM inversion on the selected video to identify the optimal noise. Additionally, we introduce a semantic-preserving rewriter to enrich the details in the text prompt. This rewriter aims to augment the original text input by providing more comprehensive and intricate information. To allow a reasonable detail compensation and maintain semantic consistency, we propose a reference-guided rewriting approach and incorporate hybrid semantics during the denoising stage. To evaluate the effectiveness of our method, we integrate the proposed modules into two established backbones and conduct extensive experiments using the widely used WebVid-10M dataset. The experimental results demonstrate the efficacy of our approach in enhancing text-to-video models.

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
