# APPENDIX

## 1 DETAILS OF SCVIDEO

**Architecture.** Figure 5 (a) illustrates the architecture of SCVideo, which follows the latent diffusion paradigm and is extended from Stable Diffusion 1.4. Roughly, SCVideo comprises three parts: 1) VQ-VAE (Razavi et al., 2019) encodes a 16*256*256 video into latent space and decodes a 4*16*32*32 latent feature map to a 16*256*256 video, allowing an efficient diffusion in latent space. (b) Text encoder CLIP (Radford et al., 2021) extracts features from the input text as conditions. (c) Sparse Spatial-Temporal attention network for denoising, which is built upon the Sparse Spatial-Temporal (SS-T) Attention. Figure 5 (b) shows the detailed architecture of SS-T attention, the input feature subsequently passes through a conv2D and conv3D, sparse causal attention (SCAttn)(Wu et al., 2022), spatial cross attention, and two temporal attentions. Hereinafter, we elaborate on these three types of attention blocks.

**Sparse Causal Attention.** Consider input feature $F \in \mathcal{R}^{bcfhw}$, as mentioned in Tune-a-video (Wu et al., 2022), sparse causal attention(SC attention) evolves from spatial self-attention (Vaswani et al., 2017) $Attention(Q, K, V) = Softmax(\frac{QK}{\sqrt{d}}) \cdot V$, with

$$Q = W^Q F', V = W^V F', K = W^K F', \tag{1}$$

where $F' = RS(F; (bf), (hw), c)$ is reshape operation that converts $F$ into shape $(bf) \times (hw) \times c$. Differently, SC attention calculates $K$ and $V$ from the first frame and the former frame.

$$Q = W^Q f_i, V = W^V[f_1; f_{i-1}], K = W^K[f_1; f_{i-1}] \tag{2}$$

where $[\cdot; \cdot]$ denotes concatenation operation, and $f_i$ is the i-th feature of $F'$ in temporal dimension.

**Spatial Cross Attention.** This module is used to find the relationship between text and image and use it to guide generation. Given the input feature $F \in \mathcal{R}^{bcfhw}$, the query, key, and value of this attention module are as follows,

$$Q = W^Q F', V = W^V \mathcal{C}, K = W^K \mathcal{C} \tag{3}$$

where $\mathcal{C}$ denotes the embedding of the text.

**Temporal Attention.** Temporal attention is similar to the spatial self-attention. The difference lies in that temporal attention is conducted on the temporal dimension:

$$Q = W^Q F'', V = W^V F'', K = W^K F'', \tag{4}$$

where $F'' = RS(F; (bhw), f, c)$.

**Training.** The parameters shared with Stable Diffusion (SD) 1.4 are directly initialized using the pretrained weights of SD 1.4. We follow Make-A-Video (Singer et al., 2022) to initialize the newly introduced parameters, the weights and bias are initialized with identity matrix and zero, respectively. This model is trained on WebVid-10M dataset, all videos are center-cropped and resized to 256*256. In addition, We take an image every 8 frames and take a total of 16 frames for each video. We only optimize 3D temporal convolution and all the attention modules during training. Our model is trained with 4 Nvidia A100-80G GPUS with batch size 5, and the total number of training steps per GPU is 80K. The learning rate is set as $3 \times 10^{-5}$ with $10^{-2}$ weight decay. We train our model using AdamW optimizer, and the hyperparameters are configured as $\beta_1 = 0.9$, $\beta_2 = 0.999$ and $\epsilon = 10^{-8}$. In addition, a cosine schedule with a linear warmup of 1000 steps is adopted during the training process.

**Sampling.** We use DDIM sampler to sample with 50 steps to synthesize videos, and the classifier-free guidance scale is set as 10.

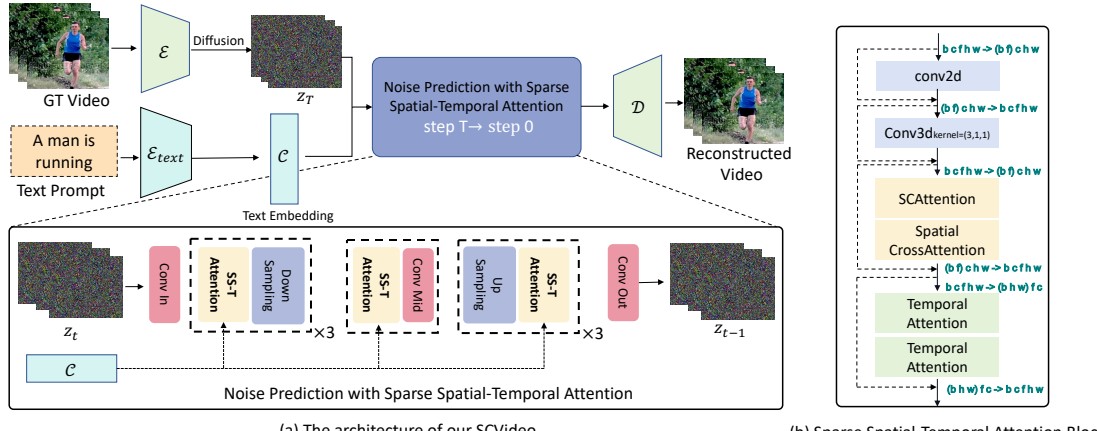

Figure 5: **Architecture of SCVideo**. Subfigure (a) illustrates the overall framework of SCVideo, subfigure (b) depicts the key module of the denoising network.

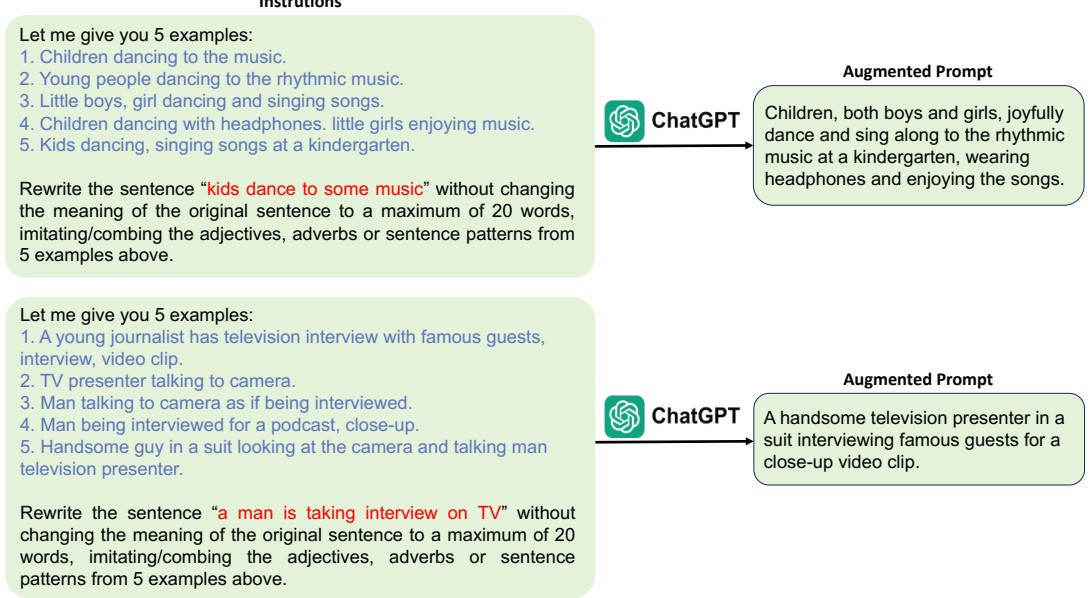

Figure 6: Examples of rewriting the text prompts using ChatGPT.

## 2 REWRITING EXAMPLES.

Figure 6 displays two rewriting examples, we can observe that the new sentences are improved by giving more modifying words, providing more effective guidance to exert the potential of the text-to-video models.

## 3 MORE QUALITATIVE RESULTS.

We also give more qualitative comparison for video generation (Figure 7) and image generation (Figure 8). From Figure 7, we can observe that the models enhanced by our modules can generate higher frame quality and better temporal consistency. As shown in Figure 8, our method also works well for image generation.

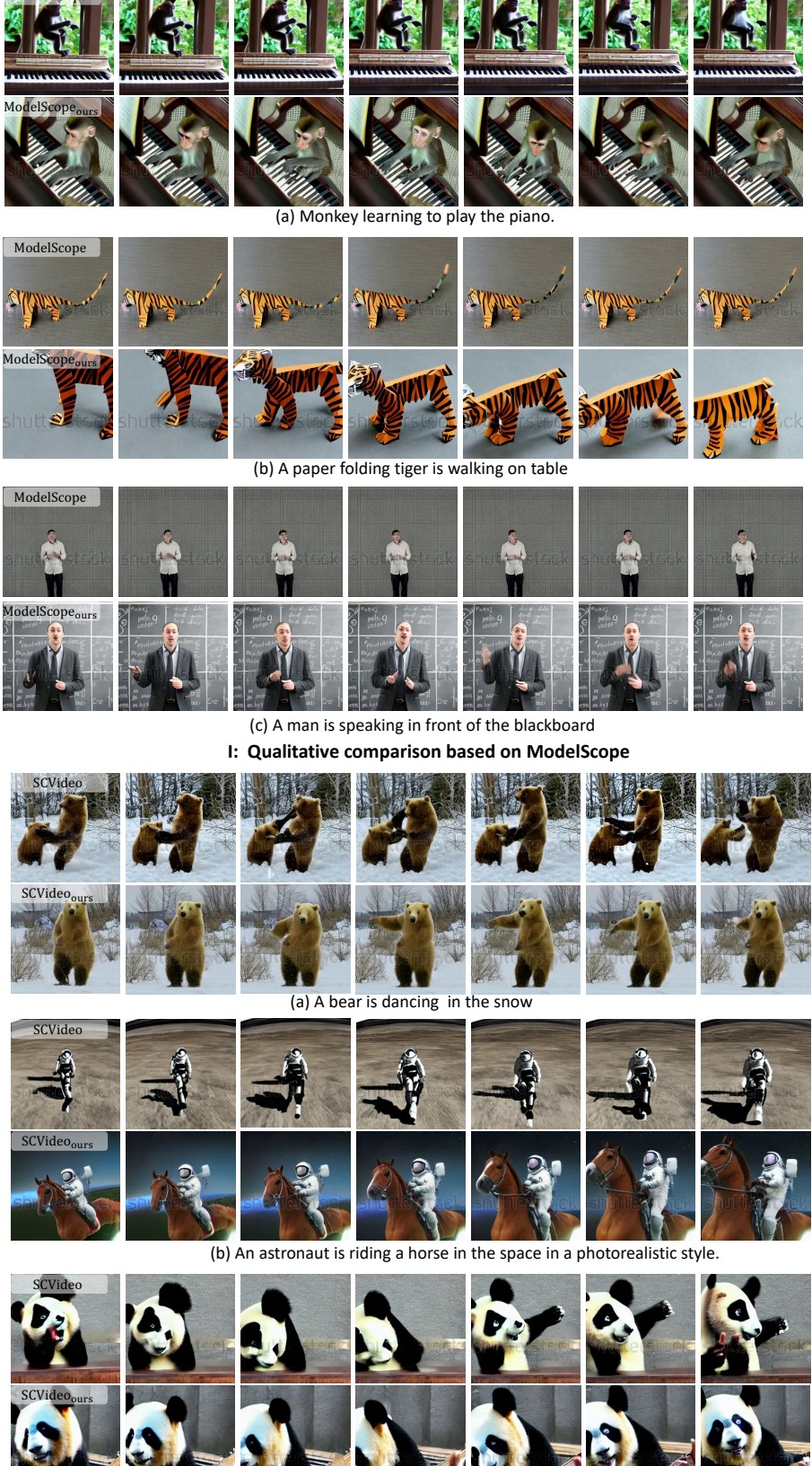

(a) Monkey learning to play the piano.

(b) A paper folding tiger is walking on table

(c) A man is speaking in front of the blackboard

**I: Qualitative comparison based on ModelScope**

(a) A bear is dancing in the snow

(b) An astronaut is riding a horse in the space in a photorealistic style.

(c) A panda is playing guitar in the bar

**II: Qualitative comparison based on SCVideo**

Figure 7: More qualitative results for video generation.

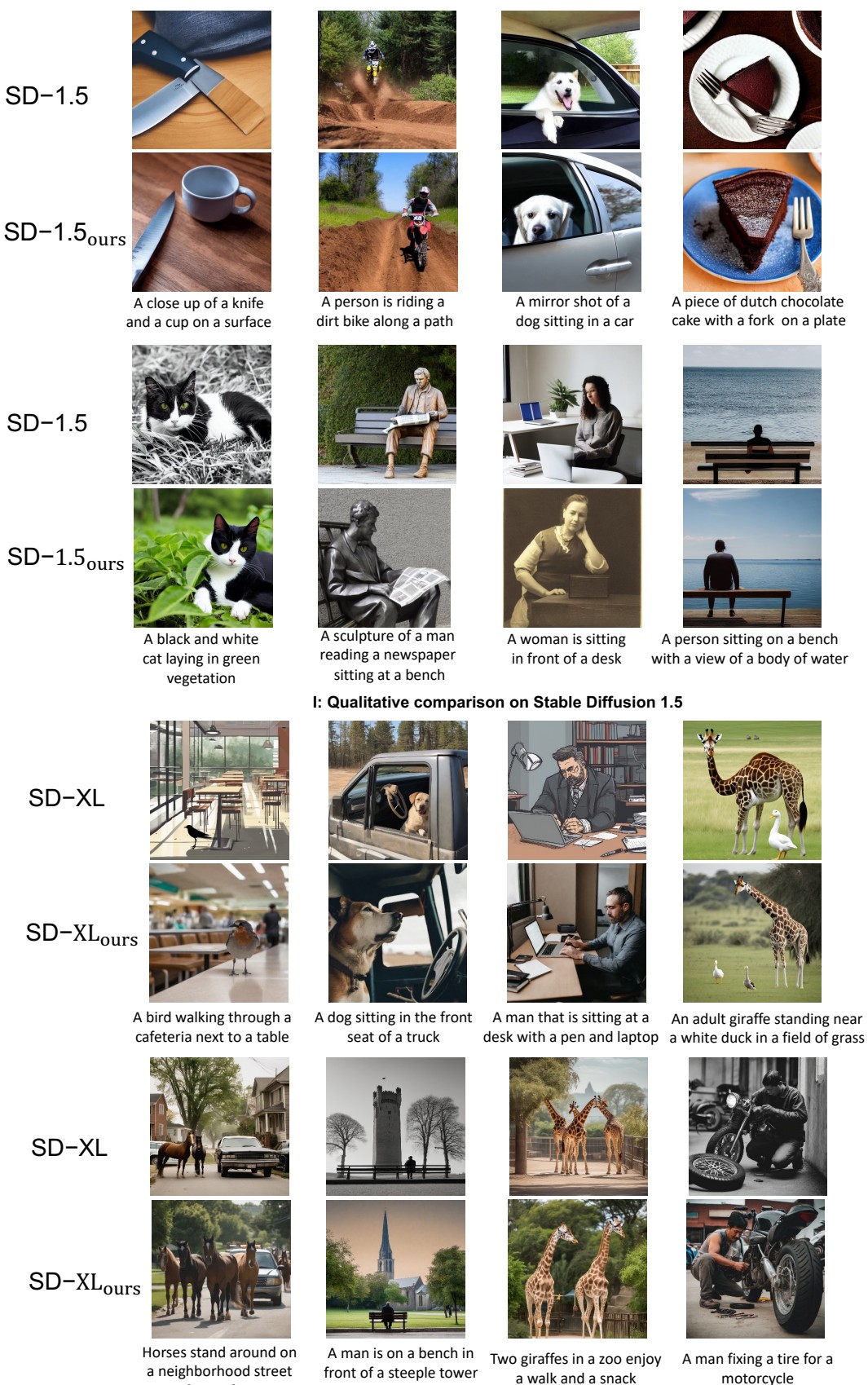

Figure 8: More qualitative results for image generation.