# OpenReview forum: "Optimal Noise Pursuit for Augmenting Text-to-Video Generation"
_ICLR.cc/2024/Conference — ICLR 2024 Conference Withdrawn Submission_

### Official Review · Reviewer_fHMH · 2023-10-23

**Soundness:** 2 fair
**Presentation:** 2 fair
**Contribution:** 1 poor
**Rating:** 3
**Confidence:** 5

**Summary:**

This paper observes that noises impact the produced videos of text-to-video models and posits that there exists an optimal noise matched to each textual input. To approximate the optimal noise, it first searches for a video closely related to the text prompts and then inverts the video to find the approximately optimal noise. In addition, it proposes a semantic-preserving rewriter to enrich the text prompt. The experiments are conducted on MSR-VTT and UCF101 using WebVid-10M as the video pool.

**Strengths:**

- The noise in the inference is indeed related to the quality of videos generated by diffusion-based text-to-video models. Exploring sampling or generating better noise is reasonable and straightforward.
- The paper is easy to understand.

**Weaknesses:**

- Obtaining a reference video through video retrieval and generating videos using the inversion noise from the reference video could result in a lack of diversity in video generation. Even though the diversity of individual frame content can be enhanced through LLMs, the crucial diversity of motion in text-to-video is constrained by inversion noise.
- Temporal coherence is a crucial aspect of video generation. However, the evaluation metrics such as FID, CLIP_FID, and CLIP-SMI are not related to the temporal coherence of videos, and the supplementary materials also do not provide the generated videos. Authors are suggested to provide the retrieved reference videos and their corresponding generated videos, along with an evaluation of their temporal coherence.
- The employed text-to-video model has been trained on WebVid-10M, and reference videos are retrieved from WebVid-10M. What happens if the retrieved video is not from WebVid-10M? Because the model has been trained on WebVid-10M, reference videos from WebVid-10M naturally invert noise that conforms to the data distribution and domain. This gives the model an inherent advantage in terms of image generation quality, and as a result, the comparison may not be entirely fair.
- Based on my understanding, the method proposed in the paper focuses on improving the quality of single-frame generation and does not have additional designs specifically aimed at enhancing temporal coherence. Therefore, it would be more appropriate to compare it with methods used to improve generation quality in text-to-image diffusion models.
- The paper lacks substantial technological contributions and seems to be primarily an integration of existing methods.
    - The inversion of a reference video is used by video editing models, such as [tune a video]， [video p2p]，[Fate Zero], and so on.
    - The weighted combination of random noise and reference noise is used in video generation techniques, as seen in various methods such as [Preserve Your Own Correlation: A Noise Prior for Video Diffusion Models] and [Free-Bloom].
    - Utilizing LLMs to enhance the content of video generation is not a new approach and has already been explored by previous works such as [MovieFactory] and [DirecT2V].

**Questions:**

Please refer to the weaknesses for details.

---

### Official Review · Reviewer_9gsF · 2023-10-29

**Soundness:** 2 fair
**Presentation:** 3 good
**Contribution:** 2 fair
**Rating:** 5
**Confidence:** 3

**Summary:**

This paper proposed two modules to enhance text=to-video quality:
Optimal Noise Approximator (ONA): Existing video diffusion models sample random noise during inference, which can result in inconsistent video quality. This method searches for a 'neighboring' video with paired text semantically similar to the input text, inverts it to get an 'optimal noise', and mixes this with random noise as the input to the model.
Semantic-Preserving Rewriter (SPR): Rewriting the input text with more details using LLMs can improve video quality, but may change semantics. To solve this, SPR performs reference-guided rewriting while introducing original text during inference for semantic consistency.


Experiments on MSR-VTT and UCF101 show ONA and SPR improve existing models like in metrics like FID, IS, FVD.

**Strengths:**

1. The paper is easy to follow and the motivation is clearly explained.
2. The methods are optimization-free and model-agnostic. The ONA and SPR modules can be integrated into any existing diffusion model without needing to re-train or modify the base model. In addition, No extra training or fine-tuning is needed to benefit from ONA and SPR.
3. Ablation studies are solid. They show the effectiveness of each proposed module (ONA, SPR) and the hyper-parameters that may affect the final performance.

**Weaknesses:**

1. As also mentioned by the authors, the computational overhead and storage needs currently limits the method’s future potential to be adopted to applications.
2. One thing I was wondering is that - - Currently the reference text-video pool consists of only real-world videos. But one may want to generate “unreal” videos using prompts like “airplane flaps its wings like a bird”. How can the retrieval process searches reasonable matches in these cases? How the performance will change in such situations?
3. Will the ONA module still be effective if the retrieval recall rate is under certain threshold? The authors may study how the performance will change if the retrieval module randomly select reference video.
4. The novelty of SPR module is limited. As also presented in related works, previous works also points out the importance of prompt rewriting.

**Questions:**

Please address my concerns listed in Weakness 2,3,4.

---

### Official Review · Reviewer_RVQz · 2023-10-31

**Soundness:** 3 good
**Presentation:** 3 good
**Contribution:** 3 good
**Rating:** 5
**Confidence:** 3

**Summary:**

The paper tackles the task of text-to-video generation and proposes to ideas to improve the existing text-to-video generation models:

Optimal Noise Approximator (ONA): Approximates the optimal noise for a given text prompt by first searching for a semantically similar video, inverting it to get a noise vector, and mixing that noise with random noise. This helps generate higher quality and more consistent videos.
Semantic-Preserving Rewriter (SPR): Rewrites the text prompt to add more details, while preserving semantic consistency with the original text.

**Strengths:**

The paper addresses an important problem in text-to-video generation - improving stability and quality by better initialising the noise vector and enriching the text prompt.

The techniques are model-agnostic and do not require architecture changes or retraining, making them easy to apply.

Experiments on two benchmarks demonstrate improvements.

**Weaknesses:**

The paper introduces two distinct methods—Optimal Noise Approximator (ONA) and Semantic-Preserving Rewriter (SPR). The two ideas, although contributing to the same overall objective, appear to be laid out in parallel, without a significant integration or interconnection between them. They are presented as separate entities.

The proposed improvements are applied to only two video generation methods, one of which is a closed-source method that the authors have in-house and we don't have many details about it. The paper would benefit for applying the proposed method to additional text-to-video generation methods to strengthen the contribution that the proposed methods are model-agnostic.

User studies assessing the quality of the generated videos could supplement automated metrics.

The idea of using a LLM for improving the text prompt it's not entirely novel and have been used for various other works, especially in the text-to-image generation [1,2].

Method is only compared against the baselines: one in-house baseline and ModelScope/VideoFussion. Qualitative or quantitative comparison with other video generation methods [3,4 etc] is lacking.


[1] Lian, Long, et al. "LLM-grounded Diffusion: Enhancing Prompt Understanding of Text-to-Image Diffusion Models with Large Language Models." arXiv preprint arXiv:2305.13655 (2023).

[2] https://cdn.openai.com/papers/DALL_E_3_System_Card.pdf

[3] Wu, Jay Zhangjie, et al. "Tune-a-video: One-shot tuning of image diffusion models for text-to-video generation." Proceedings of the IEEE/CVF International Conference on Computer Vision. 2023.

[4] Singer, Uriel, et al. "Make-a-video: Text-to-video generation without text-video data." arXiv preprint arXiv:2209.14792 (2022).

**Questions:**

see above

---

### Official Review · Reviewer_y9ER · 2023-11-01

**Soundness:** 2 fair
**Presentation:** 3 good
**Contribution:** 2 fair
**Rating:** 5
**Confidence:** 5

**Summary:**

This paper proposed an optimization-free, model-agnostic approach for enhancing text-to-video generation. It proposed an optimal noise approximator (ONA) and a semantic-preserving rewriter (SPR) as two main contributions. ONA focuses on approaching the optimal noise by combining the retrived inverted-DDIM noise and Gaussian noise for the given text prompt. SPR leverages ChatGPT to rephrase the input prompt for more details. Experiments show that on two pre-trained text-to-video models, ModelScope and SCVideo, using the proposed approach is able to improve the results qualitatively and quantitatively.

**Strengths:**

1. The paper is well-written and easy to follow.
2. The motivation for this paper is interesting. It aims to improve the T2V quality without fine-tuning the pre-trained model.
3. Table 1 and Figure 4 show that the generated results can be improved qualitatively and quantitatively by using the proposed results

**Weaknesses:**

1. The authors claimed that the unsatisfiring quality of generated results could be random input noises. However, I didn't find enough analysis and discussion in the paper to prove such claims. Why can random noise not always generate high-quality results? It is because the learned T2V model is not good enough?
2. How to maintain the creativity of the original text-to-video model if we retrieve a noise? Will such operation reduce the diversity and generalizability of the original model?
3. I didn't find any videos in the supplementary material. It is hard to tell wether the generated quality especially temporal consistency is improved from still images. I suggest the authors provide some demo videos.
4. Using retrieval* to improve the quality of input noise is not novel, nor using LLM to improve the input prompt. I hope the authors could clarify this paper's contributions

* Blattmann et al., Semi-Parametric Neural Image Synthesis, NeurIPS 2022

**Questions:**

see weaknesses